# Predicting Visual Acuity Deterioration and Radiation-Induced Toxicities after Brachytherapy for Choroidal Melanomas

**DOI:** 10.3390/cancers11081124

**Published:** 2019-08-06

**Authors:** Charlotte A. Espensen, Ane L. Appelt, Lotte S. Fog, Anita B. Gothelf, Juliette Thariat, Jens F. Kiilgaard

**Affiliations:** 1Department of Oncology, Section of Radiotherapy, Copenhagen University Hospital, Rigshospitalet, 2100 Copenhagen, Denmark; 2Department of Ophthalmology, Copenhagen University Hospital, Rigshospitalet, 2100 Copenhagen, Denmark; 3Leeds Institute of Medical Research at St James’s, University of Leeds, Leeds LS9 7TF, UK; 4Department of Physical Sciences, The Peter MacCallum Cancer Centre, Melbourne 3000, Australia; 5Department of Radiation Oncology, Centre Francois Baclesse, 14000 Caen, France; 6Laboratoire de Physique Corpusculaire IN2P3/ENSICAEN, 14000 Caen, France; 7Laboratoire de Physique Corpusculaire IN2P3/ENSICAEN-UMR6534, Unicaen–Normandy University, 14000 Caen, France

**Keywords:** choroidal melanoma, brachytherapy, normal tissue complication probability, dose-response

## Abstract

Ruthenium-106 (Ru-106) brachytherapy is an established modality for eye-preserving treatment of choroidal melanoma. To achieve optimal treatment outcomes, there should be a balance between tumour control and the risk of healthy tissue toxicity. In this retrospective study, we examined normal tissue complication probability (NTCP) for visual acuity deterioration and late complications to aid the understanding of dose-dependence after Ru-106 treatments. We considered consecutive patients diagnosed with choroidal melanoma and primarily treated at a single institution from 2005–2014. Treatment plans were retrospectively recreated using dedicated software and image guidance to contour the tumour and determine the actual plaque position. Dose distributions were extracted from each plan for all relevant anatomical structures. We considered visual acuity deterioration and late complications (maculopathy, optic neuropathy, ocular hypertension, vascular obliteration, cataract and retinal detachment). Lasso statistics were used to select the most important variables for each analysis. Outcomes were related to dose and clinical characteristics using multivariate Cox regressions analysis. In total, 227 patients were considered and 226 of those were eligible for analysis. Median potential follow-up time was 5.0 years (95% CI: 4.5–6.0). Visual acuity deterioration was related to optic disc-tumour distance and dose metrics from the retina and the macula, with retina V10Gy showing the strongest correlation. Macula V10Gy was the only dose metric impacting risk of maculopathy, while optic disc-tumour distance also proved important. Optic disc V50Gy had the largest impact on optic neuropathy along with optic disc-tumour distance. Optic disc V20Gy was the only variable associated with vascular obliteration. Lens D2% had the largest impact on the risk of cataract along with older age and the largest base dimension. We found no variables associated with the risk of ocular hypertension and retinal detachment. Visual acuity deterioration and most late complications demonstrated dependence on dose delivered to healthy structures in the eye after Ru-106 brachytherapy for choroidal melanomas.

## 1. Introduction

Ruthenium-106 (Ru-106) brachytherapy is an established eye-preserving treatment modality for choroidal melanoma, with good local tumour control rates [1,2]. However, an optimal treatment strategy has to balance tumour control and the risk of healthy tissue toxicity, especially when a major treatment aim is to preserve normal function. We established a dose–response relationship for tumour control probability (TCP) after ruthenium-106 (Ru-106) brachytherapy in a previous work [3], but corresponding models for normal tissue complication probabilities (NTCP) are required in order to decide on treatment modality and optimise treatment at an individual patient level. We examined the relationship between radiation dose to healthy tissues and the risk of visual acuity deterioration and radiation-induced toxicity for a large cohort of patients with choroidal melanoma treated with Ru-106 brachytherapy.

## 2. Methods and Materials

### 2.1. Patient Material

In this retrospective study, we considered consecutive patients diagnosed with primary choroidal melanoma and treated with Ru-106 brachytherapy from January 2005 to December 2014 at a single tertiary referral institution. Baseline patient, tumour, and treatment characteristics were prospectively registered for all the patients in a local database. Clinical outcome data were retrospectively reviewed from patient records along with evaluation based on imaging material. The acquired data were stored in a dedicated database approved by the Danish Data Protection Agency (ref 2016-41-4897) and the Danish Health Authority (ref 3-3013-980/1/). Patients were excluded from the analysis if treatment records and follow-up records or continuously recorded retinographies were unavailable. The patient selection process is illustrated in Figure 1.

### 2.2. The Treatment Procedure

Non-invasive examination of the retina was performed before treatment recommendations were made. Ophthalmoscopy, fluorescein angiography, and/or optical coherence tomography (OCT) was used along with ultrasound B-scans to assess tumour extension. Patients were referred to brachytherapy if they had locally confined disease and the tumour dimensions were within the limits treatable with Ru-106 plaques (5 mm in height) [4]. Each treatment was performed in the operating theatre, and the Ru-106 plaque was surgically sutured onto the sclera adjacent to the tumour. Ru-106 plaques were manufactured and provided by Eckert and Ziegler (BEBIG GmbH, Berlin, Germany). Plaques with different sizes and shapes were available (CCA, CCB, CCC, and COB), and the most suitable was chosen according to tumour location within the eye and the size of the tumour. A margin of 2 mm from the tumour to the plaque borders was preferred, but an eccentrically located plaque was used for some cases [5]. Correct positioning of the plaque was ensured with ultrasound directly after plaque placement and one day post-surgery. Three experienced onco-ophthalmologists performed the treatments over the 10-year period. Tumour height and base dimensions were measured using ultrasound as part of the treatment procedure. An apical dose of 100 Gy was prescribed for all the treatments, and the treatment times were calculated using an in-house developed spreadsheet accounting for the activity of the plaque at the insertion time and height of the tumour.

### 2.3. Regular Assessment

Patients were followed regularly by an onco-ophthalmologist consultant every third month during the first year, every sixth months during the second year, and annually thereafter for at least five years, if possible. Clinical outcomes including visual acuity and radiation-induced toxicities were evaluated at each visit.

### 2.4. Dose Distribution Analysis

Each treatment was retrospectively recreated using the three-dimensional (3D) image-guided treatment planning software Plaque Simulator (version 6.5.9, EyePhysics, LLC, Los Alamitos, CA, USA). The tumour volume was based on contouring of the tumour base on pre-treatment retinographies and using the ultrasound measure for tumour height. The plaque location was estimated from the radiation scar on post-treatment retinographies, or alternatively based on the surgery note from the patient record. Identification of the macula and the optic disc on the retinography was done manually, and was crucial to correctly calibrate the image to map the eye model. They were both contoured as circular structures with diameters of 1.5 mm and 2 mm, respectively. A standard eye size was used with anterior–posterior diameter of 26.2 mm and an equatorial diameter of 24.0 mm. The lens was outlined as a volume of size 10 × 4 mm (diameter × thickness), while the retina was outlined as a structure from the posterior pole to the limbus with a 1-mm inset from the outer surface of the sclera [6]. Each plan was recreated in close collaboration with an experienced onco-ophthalmology consultant. Full 3D dose distributions were calculated based on information regarding the insertion and removal times of the plaque. Complete dose area histograms were extracted for the macula, the retina, and the optic disc, while dose volume histograms were extracted for the globe and the lens.

### 2.5. Definition of Outcomes

Table 1 lists the clinical findings and equipment used in the examination of each of the late complications.

We performed two visual acuity analyses: the first including the full cohort (group 1), and the second including solely patients with a pre-treatment visual acuity ≤0.5 logMAR (logarithm of minimum angle of resolution) (the limit for public blindness and the minimum driving license requirement, group 2).

### 2.6. Data Analysis

For each late complication, a consultant ophthalmologist with several years’ experience in ocular oncology pre-specified clinical factors as well as relevant normal tissue structures for which to extract dose metrics. This resulted in more than 30 potential variables per late complication to include in the analysis. See details in the analysis plan in the Appendix A, including all the explanatory factors considered for each endpoint.

Variable selection was performed using Lasso statistics to identify and eliminate non-informative variables with poor association with the specific late complication. Ten-fold cross-validation was used to estimate the optimal shrinkage parameter (λ). The optimal value of λ was the value that minimised the prediction error from the cross-validation; however, it had numerous redundant predictors. Thus, to achieve the simplest model, we used a larger value of λ, where the error remained within one standard error of the minimum.

Dose–response relationships were evaluated from Cox proportional hazards regression. The time to event was measured from the start of treatment to whichever occurred first: late complication, re-treatment (due to recurrence), censoring (due to competing events), death, or end of follow-up (May 2019). For maculopathy, tumours located under the macula at diagnosis were accounted for in the baseline characteristics, whereas tumour growth involving the macular region at any time during follow-up was censored at first presentation. Additionally, for optic neuropathy, tumours partly overlapping the optic disc at diagnosis were accounted for in the baseline characteristics, whereas tumour growth overlapping the optic disc at any time during follow-up was censored at first presentation. The Danish Health care system did not reimburse preventive treatment of ocular complications; therefore, patients only received treatment for late complications after the complications occurred. Consequently, the time-to-event was not affected. In few cases with lens removal during primary surgery, the patients were censored at that time. See the analysis plan in the Appendix A for details on competing events for each specific late complication. The reverse Kaplan–Meier method was used to estimate the potential follow-up time [9].

Visualisation of the dose–response relationship was conducted by plotting the Cox regression five-year risk estimates as a function of dose with all the other model variables kept at a constant value (typically the median value of the cohort, e.g., median tumour height).

We assessed model performance using the concordance index and the Brier score. Furthermore, Schoenfeld residuals were used to evaluate the proportional hazard assumption on the time independency of the variables.

For explorative purposes, we conducted additional logistic regression analyses for visual acuity loss, with follow-up time as an explanatory variable. See details in the Appendix A.

## 3. Results

In total, 227 patients were treated during the 10-year period. One died immediately after treatment before any follow-up routines were performed, and was therefore excluded from the study; thus, 226 were considered for analysis. Median potential follow-up time for the remaining cohort was five years (95% CI: 4.5–6.0). Patient and tumour characteristics are listed in Table 2. In total, 50 recurrences were observed, and the overall five-year local control estimate was 78%. Overall estimates of five-year freedom from toxicity were done using Kaplan–Meier and listed in Table 2. Kaplan–Meier curves for all the complications are illustrated in the Appendix A.

### 3.1. Visual Acuity Analysis

The optimal model for visual acuity deterioration (group 1) included optic disc–tumour distance, the area of the retina receiving 10 Gy (retina A_10Gy_), and two macula dose metrics (macula A_20Gy_ and A_80Gy_). The hazard ratios indicated a strong dependency of retina A_10Gy_ (see Table 3). The relationship is illustrated in Figure 2A. Poor pre-treatment visual acuity and close proximity to the optic disc–tumour distance was also associated with an increased risk of visual acuity deterioration. The dose–response model divided into pre-treatment visual acuity of 0.2 logMAR and 1.8 logMAR is illustrated in Figure 2B.

The concordance index and Brier score showed good correlation between the observed and predicted five-year visual acuity loss; thus, the performance of the model was acceptable, as illustrated in the Appendix A. 

None of the variables considered demonstrated an association with the risk of loss of pre-treatment visual acuity (group 2).

The results from the logistic regression analyses based on pre-treatment and last visual acuity deterioration are provided in the Appendix A.

### 3.2. Late Complications

The hazard ratios (HRs) from Cox regression analyses for each of the late complications are listed in Table 3. The selection procedure did not find any relevant factors related to late toxicity for ocular hypertension or retinal detachment.

The area of macula receiving 10 Gy (macula A_10Gy_) had the largest HR for maculopathy and a considerable impact on the risk of developing this late complication. The dose–response relationship is illustrated in Figure 3A.

Decreased optic disc–tumour distance demonstrated a considerable risk for developing optic neuropathy. Furthermore, the area of the optic disc receiving 50 Gy (optic disc A_50Gy_) had a considerable impact on the risk. This dose–response relationship is illustrated in Figure 3B.

The development of vascular obliteration was strongly associated with optic disc A_20Gy_ as the only variable. This relationship is illustration in Figure 3C.

The near-maximum dose delivered to the lens (lens D_2%_) was the only dose metric associated with cataracts, although with a relatively weak correlation. The dose–response for this relationship is illustrated in Figure 3D. Age and largest base dimension were associated with the largest HRs for the risk of developing cataracts.

The concordance indices and Brier scores indicated good performance for all the above models. See the Appendix A. Median dose area/volume histograms for each structure are illustrated in the Appendix A.

## 4. Discussion

In this retrospective analysis, we demonstrated clear relationships between specific dose metrics for healthy tissues and the risk of late complications following Ru-106 brachytherapy. Radiation dose dependence was found for some of the endpoints (visual acuity deterioration, maculopathy, optic neuropathy, vascular obliteration, and cataracts) but not for others (ocular hypertension and retinal detachment). Clinical factors (optic disc–tumour distance, age at treatment, largest base dimension, and pre-treatment visual acuity) also correlated with outcome for some endpoints. Detailed dose analyses were performed, utilising full dose area/volume histograms extracted from retrospectively recreated treatment plans with 3D image guidance.

We found a strong correlation between the risk of visual acuity deterioration and the area of retina receiving 10 Gy (retina A_10Gy_), with a 50% risk of visual acuity deterioration when 20% of the retina received 10 Gy. Furthermore, the risk correlated with various macula dose metrics (macula A_20Gy_ and macula A_80Gy_). These findings were in accordance with those of Aziz et al., who demonstrated total fovea dose as the most significant variable associated with an increased risk of visual acuity loss in 311 patients [10]. Additionally, Heilemann et al. investigated visual acuity loss in 45 patients and found retina D_2%_ as the main risk factor [11]. However, neither of them systematically explored a wide range of dose metrics.

Previous works have identified specific clinical variables as predictors for visual acuity loss, but reports are diverse and possibly contradictory, emphasising the complex mechanisms of visual acuity deterioration following Ru-106 treatments. Thus, the specific underlying causes remain not fully understood, and radiation-induced visual acuity deterioration might originate from several discrete pathophysiologies. Isager et al. investigated visual outcomes for 55 patients, and found tumour height and the largest base dimension as the most important risk factors for visual acuity deterioration, but they did not perform multivariate analyses, nor did they include dose in their analysis [12]. According to Bergman et al., initial visual acuity was main risk factor along with the distance from the tumour to the fovea in a study with 579 patients. However, they did not consider dose either [13]. Additionally, Damato et al. reported on 458 patients and also found initial visual acuity and tumour location as risk factors for visual acuity deterioration [5]. These findings were in line with the present study, in which optic disc–tumour distance and pre-treatment visual acuity were important clinical predictors for visual acuity deterioration. However, dose was, contrary to our results, not significant in the multivariate analysis by Damato et al.

Risk factors for specific late complications have also been previously reported. Tagliaferri et al. recently found tumour location as the strongest risk factor for radiation-induced maculopathy based on a study with 197 patients [14]. They did not find any significance of dose in multivariate analysis. However, the location of the tumour might be strongly correlated with the macular dose, and thereby have worked as a surrogate for the effect of dose in the multivariate analysis. Summanen et al. found a 30% five-year risk of developing radiation-induced maculopathy after Ru-106 treatments based on 100 patients [15], while Naseripour et al. reported a five-year risk of maculopathy of 20% based on 51 patients [16]. This is slightly less than the 45% five-year risk found in our work. This could possibly be explained by a lack of consensus in the reporting of maculopathy. Finger et al. have established guidelines for general strategies in the reporting of retinopathy [17]. Extension to other radiation-induced late complications remains highly needed.

The reported risk of radiation-induced optic neuropathy varies considerable throughout the literature. According to Naseripour et al., the five-year risk was 40% (compared to 32% in our series), but no variables were significantly associated with the risk of developing optic neuropathy in their multivariate analysis. Summanen et al. reported a five-year risk of 12% and according to their study, the optic disc–tumour distance was the strongest predictor in radiation-induced optic neuropathy. This corresponds to the findings in our study. Furthermore, we found various optic disc metrics (optic disc V_50Gy_ and V_20Gy_) as important risk factors for radiation-induced optic neuropathy.

Post-treatment vascular obliteration is to some extent expected after treatments with brachytherapy, especially for vessels near the tumour and thus near the plaque. However, only a limited number of studies describe predictors for vascular obliteration. Rouberol et al. found that the risk decreased with age and anterior location, but they used a prescription scheme that deviated from current standard regimens [18]. We found optic disc V_20Gy_ as the only variable associated with increased risk.

Summanen et al. reported a five-year risk of radiation induced cataract of 37% [15]. They found tumour height as the strongest risk predictor. Naseripour et al. demonstrated a five-year risk of 38%, but did not report any prognostic factors [16]. The lens has previously been reported as a fairly radiosensitive structure [19]. However, it should be kept in mind that cataracts are often a treatable condition, and sparing of the lens is thus less essential in the treatment optimisation.

The effect of dose on post-treatment retinal detachment has been demonstrated in a recent study by Heilemann et al. They found a strong correlation with both retina D_2%_ and D_mean_. However, our study did not find any support for this. Retinal detachment was the least frequent late complication in our series. Several patients had retinal detachment at the time of diagnosis as a result of tumour exudation. While exudation diminishes as the tumour shrinkages due to Ru-106 brachytherapy [20], the retina fastens accordingly.

Published data elucidating the effect, if any, of dose rate on tumour control probability and late complications after Ru-106 brachytherapy is limited. According to Naseripour et al., a low dose rate did not have any significant negative effect on local tumour control [16]. Furthermore, Fili et al. found that dose rate was not associated with the risk of secondary enucleation. However, they suggested that longer treatment times (and thus a lower dose rate) might be associated with increased ocular irritation [21]. According to Mossböck et al., a high dose rate (as seen with shorter treatment times) might furthermore be associated with increased late complications and visual acuity deterioration [22]. We included overall treatment time in the selection process to account for any dose rate effect, but we found no association with any of the late complications.

We only examined a single plaque source in the current study. However, compared to iodine-125 (I-125), Ru-106 generally causes fewer late complications [23,24]. Miguel et al. found the type of plaque to be an important risk factor for optic neuropathy after I-125 brachytherapy, along with age, the largest base dimension of the tumour, and dose to the optic nerve [25]. Furthermore, they found a 43% five-year risk of retinal detachment, with patient age at treatment and size of the plaque as important risk factors. Thus, plaque types utilising different isotopes might see different dose dependence than what has been observed here.

We estimated the dose delivered to the specific structures based on full dose area/volume histograms extracted from retrospectively recreated treatment plans. This approach used post-treatment retinographies to assess the actual plaque position (and hence the actually delivered dose) rather than an ideal pre-treatment plan. To the best of our knowledge, no previous studies have used 3D image guidance along with relevant clinical variables to perform normal tissue complication probabilities for choroidal melanoma patients.

Lasso statistics were used for variable selection, allowing for the consideration of a wide range of possible predictors; non-informative variables were excluded, while the appropriate predictors associated with the risk of each specific endpoint were included in the subsequent Cox regression analysis. This is a well-established method to manage large numbers of potential explanatory variables, but it is important to recognise that it tends to have difficulties when factors are strongly correlated [26]. Even though the cohort size was relatively large, some of the late toxicities had a limited number of events (retinal detachment and ocular hypertension). This could potentially explain why the selection process was unsuccessful for these late complications. Furthermore, it is important to recognise that the retrospective nature of the treatment planning and the toxicity scoring could potentially introduce bias.

Prescription procedures for Ru-106 treatments remain controversial [20]. Some centres use an apical prescribed dose of 85 Gy [1,24], some use 100 Gy to the apex [2,13], while others use at least 130 Gy to the apex along with restricted tumour base doses of at least 700 Gy [27]. Eccentrically located plaques are also reported frequently in the literature, and studies have indicated that this strategy can spare healthy tissue in some cases [5]. Further studies have suggested that eccentric positioning can lead to underdosage of the tumour [28]. We recently reported that the minimum dose to the tumour is crucial for local control (Espensen et al., submitted manuscript(companion paper)), and compromises on tumour coverage should be made with caution and assessed relative to the potential benefits of visual acuity preservation and reduced probability for late complications.

## 5. Conclusions

This study demonstrates the presence of dose–response relationships for late complications after Ru-106 brachytherapy for choroidal melanoma. Specific dose metrics were important for distinctive late complications. Dose–response models were established for each late complication, thereby enabling potential for optimising clinical outcomes by personalising dose prescriptions and conducting treatment optimisation.

## Figures and Tables

**Figure 1 cancers-11-01124-f001:**
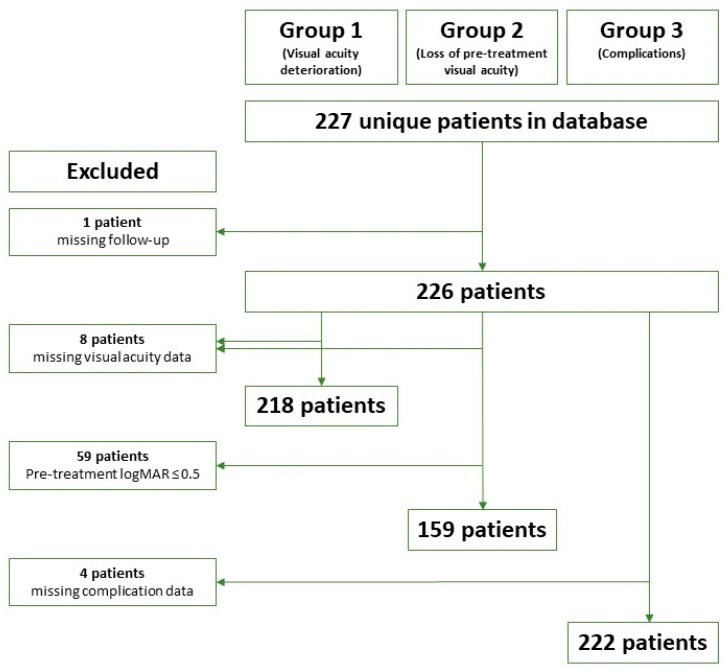
The patient selection process. LogMAR: logarithm of minimum angle of resolution.

**Figure 2 cancers-11-01124-f002:**
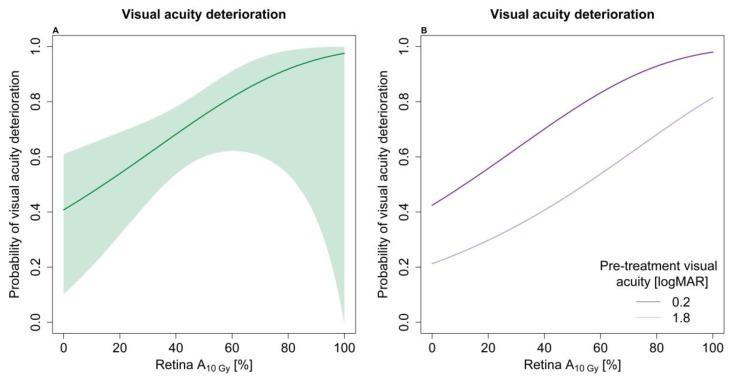
(**A**) Dose–response of visual acuity deterioration as a function of the area of the retina receiving 10 Gy (retina A_10Gy_). The model adjusts for optic disc–tumour distance (2.4 mm), pre-treatment visual acuity (0.3 logMAR), macula V_20Gy_ (89%), and macula A_80Gy_ (8%). The shaded area indicates the 95% confidence intervals. (**B**) Dose–response of visual acuity deterioration as a function of retina A_10Gy_ for two pre-treatment visual acuity measures (0.2 and 1.8 logMAR); all the other factors were kept as for Figure 2A.

**Figure 3 cancers-11-01124-f003:**
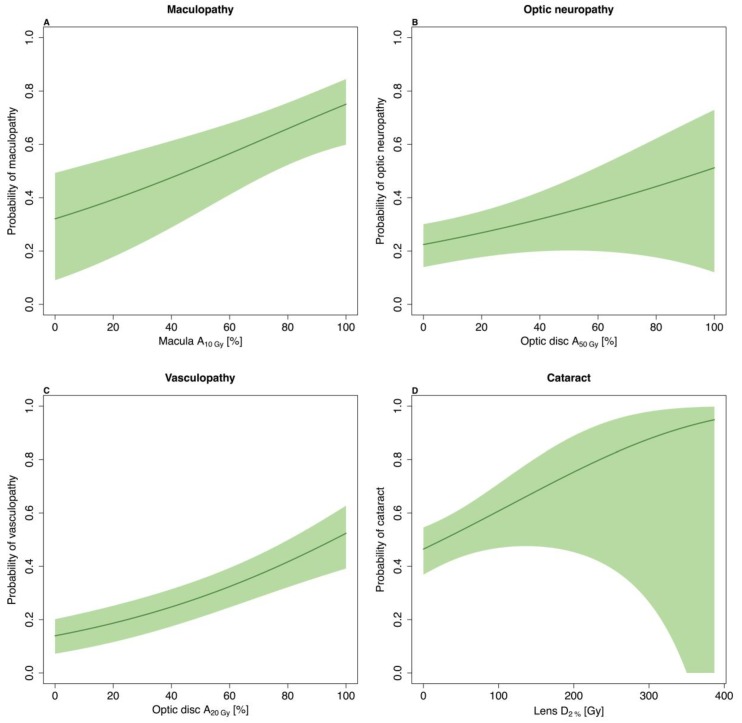
(**A**) Dose–response of maculopathy as a function of the area of macula receiving 10 Gy (macula A_10Gy_). The model adjusts for optic disc–tumour distance (2.4 mm). (**B**) Dose–response of optic neuropathy as a function of optic disc A_50Gy_. The model adjusts for optic disc–tumour distance and optic disc A_50Gy_ (39.4 Gy). (**C**) Dose–response of vasculopathy as a function of optic disc A_20Gy_. (**D**) Dose–response of cataract as a function of lens D_2%_. The model adjusts for age at treatment (62 years) and largest base dimension (11.3 mm). The shaded areas represent the 95% confidence interval of the risk estimates.

**Table 1 cancers-11-01124-t001:** List of the considered late complications, the clinical findings for each, and the equipment used in the examination. OCT: optical coherence tomography, logMAR: the logarithm of minimum angle of resolution.

Late Complication	Clinical Findings	Important Examination
Visual acuity deterioration	Increase of minimum 0.3 logMAR from the pre-treatment measure	Snellen’s chart (converted to logMAR for statistical purposes)
Maculopathy	Micro aneurysms, ischemia, oedema, and/or atrophy in the macular region [7]	Ophthalmoscopy, retinography, OCT and/or fluorescein angiography.
Optic neuropathy	Swelling, ischemia, atrophy and/or pallor occurring optic disc [8]	Ophthalmoscopy, retinography and/or OCT
Ocular hypertension	Intraocular pressure ≥21 mm Hg (at least three months post-treatment)	Tonometry
Vascular obliteration	Narrow and obliterated blood vessels on the retina	Ophthalmoscopy
Cataract	Lens opacities along with gradually deterioration of the visual acuity	Slit lamp examination or ophthalmoscopy
Retinal detachment	Fluttering membrane	Ophthalmoscopy

**Table 2 cancers-11-01124-t002:** Descriptive statistics of study participants (n = 226): Patient, tumour, and treatment characteristics, and list of late complications with raw incidence and five-year probability (based on Kaplan–Meier estimates) including 95% confidence interval (CI). logMAR: the logarithm of minimum angle of resolution. VA: Visual acuity. Median (IQR: interquartile range).

Patient Characteristics	Value (Median (IQR))	
Age (years)	62 (53–69)	
Gender male/female	118/108	
Eye left/right	117/109	
Follow-up (years)	5 (95% CI: 4.5–6.0)	
Pre-treatment VA (logMAR)	0.3 (0.0–0.6)	
Pre-treatment VA ≤0.5 logMAR (y/n)	165/61	
Last VA (logMAR)	0.9 (0.3–3.0) (NA = 4)	
Last VA ≤0.5 logMAR (y/n)	82/140 (NA = 4)	
Tumour characteristics		
Largest basal dimension (mm)	11.4 (9.0–13.3)	
Height (mm)	3.9 (2.8–5.8)	
Optic disc–tumour distance (mm)	2.4 (0.4–4.9)	
Macula–tumour distance (mm)	2.5 (0.1–5.0)	
Treatment characteristics		
Treatment (time hours)	120 (74–191)	
Plaque type CCA/CCB/CCC/COB	53/101/12/60	
**Late complication**	**Number (%)**	**5-year probability of freedom from toxicity (95% CI)**
Loss of pre-treatment visual acuity	101 (66)	29 (22–38)
Visual acuity deterioration	136 (62)	35 (29–43)
Maculopathy	64 (29)	45 (36–56)
Optic neuropathy	62 (28)	68 (62–76)
Ocular hypertension	26 (12)	87 (82–92)
Vascular obliteration	63 (28)	70 (63–77)
Cataract	103 (46)	52 (45–61)
Retinal detachment	15 (7)	94 (91–97)

**Table 3 cancers-11-01124-t003:** Hazard ratios (HR) with 95% confidence intervals (CI) for each of the late complications. logMAR: the logarithm of minimum angle of resolution. VA: Visual acuity. VA: Visual acuity.

Visual Acuity Deterioration	Hazard Ratio (95% CI)
Optic disc-tumour distance ^+^	0.91 (0.85–0.97)
Pre-treatment VA (1 logMAR increase)	0.59 (0.44–0.80)
Retina A_10Gy_ *	1.22 (1.03–1.44)
Macula A_20Gy_ *	1.04 (0.98–1.10)
Macula A_80Gy_ *	0.92 (0.62–1.38)
Loss of pre-treatment visual acuity	No variables selected
Maculopathy	
Optic disc–tumour distance ^+^	0.87 (0.79–0.96)
Macula A_10Gy_ *	1.15 (1.05–1.26)
Optic neuropathy	
Optic disc–tumour distance ^+^	0.75 (0.63–0.89)
Optic disc A_50Gy_ *	1.11 (1.02–1.22)
Optic disc A_20Gy_ *	1.08 (0.98–1.18)
Ocular hypertension (post-treatment)	No variables selected
Vascular obliteration	
Optic disc A_20Gy_ *	1.17 (1.11–1.25)
Cataract	
Age at treatment (10 years increase)	1.38 (1.17–1.62)
Largest base dimension ^+^	1.08 (1.01–1.16)
Lens D_2%_ (10 Gy increase)	1.04 (1.01–1.07)
Retinal detachment (post-treatment)	No variables selected

* 10%-point increase, ^+^ 1-mm increase.

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
