# Peer review of "Predicting Visual Acuity Deterioration and Radiation-Induced Toxicities after Brachytherapy for Choroidal Melanomas"

_cancers, 2019, doi:10.3390/cancers11081124_

Round 1
Reviewer 1 Report
Dear authors, I had the chance to review your manuscript "Predicting visual acuity deterioration and radiation-induced toxicities after brachytherapy for choroidal melanomas"
I think the paper is well written, the methods are in accordance and improve on the current standards, the analyses are sound, the discussion covers most of the aspects of the topic, and the conclusions are not over-confident.
Therefore I think the manuscript will contribute to a better understanding of this topic.
I would like to mention, that the manuscript there are some inacurracies and some facts are missing, which I would ask to to include in the final version.
In Detail
21: please explain TCP at time of first occurrence in the text
70 – 84: According to appendix A, the retina was defined as surface, the macula and the optic disc as volume using plaque simulator for DVH. What was the thickness of macular and optic disc volume? Why not using volume for the retina as well?
150: Here volume of the retina is used in stead of surface area
Appendix:
Legend Fig B2: Visual acuity 'loss'.
Fig D1: Retina is used as volume again, not surfce.
Pease stick to either upper-case or lower-case for structures
I do not have access to the companion paper.
Howerver, I miss several facts:
How many recurrences were there?
Another main point is:
How were the late complications ltreated?
If you treat one complication, this might also have an effect on the occurence of another one, and follow-up for this other risk should perhaps be censored.
e.g. Treatment of optic neuropathy with Bevacizumab might prevent / delay occurence of radiation maculopathy.
How did you deal with this problem?
277: In fact, cataract is a treatable condition. To my understanding, dose to lens has been used as a surrogate for dose to ciliary body, which is much more difficult do calculate, as outlining / defining the volume of the ciliary body, and therefore might be used to predict neovascular glaucoma.
Author Response
Dear Reviewer,
Thank you for the constructive review. We think that it has improved the quality of the manuscript.

Reviewer 2 Report
Please review Table 1. Starting with line 2, the second and third coloumn are not properly aligned and the reader dos not know which line in those coloumns belong to which line in the first one (at least as shown in my pdf-viewer).
The spelling aneurism is possible, but is much less common than aneurysm (1:20 according to wictionary). Please consider re-spelling.
Line 120: What do you mean by "cataract aphakic patients"?
Caption Figure 2: Please explain Retina V10Gy (as already done within the main text). Otherwise, the figure is not understandable
Table 3: Same problem as with table 1. The lines do not match (the error starts with the line maculopathy and the shift becomes greater till the end of the table)
Author Response

(The authors gave the same response as above.)
